# TNIK inhibition abrogates colorectal cancer stemness

Mari Masuda[1], Yuko Uno[2], Naomi Ohbayashi[3,*], Hirokazu Ohata[4,*], Ayako Mimata[1,*], Mutsuko Kukimoto-Niino[3], Hideki Moriyama[2], Shigeki Kashimoto[2], Tomoko Inoue[2], Naoko Goto[1], Koji Okamoto[4], Mikako Shirouzu[3], Masaaki Sawa[2,*] & Tesshi Yamada[1,*]

Canonical Wnt/β-catenin signalling is essential for maintaining intestinal stem cells, and its constitutive activation has been implicated in colorectal carcinogenesis. We and others have previously identified Traf2- and Nck-interacting kinase (TNIK) as an essential regulatory component of the T-cell factor-4 and β-catenin transcriptional complex. Consistent with this, *Tnik*-deficient mice are resistant to azoxymethane-induced colon tumorigenesis, and $Tnik^{-/-}/Apc^{min/+}$ mutant mice develop significantly fewer intestinal tumours. Here we report the first orally available small-molecule TNIK inhibitor, NCB-0846, having anti-Wnt activity. X-ray co-crystal structure analysis reveals that NCB-0846 binds to TNIK in an inactive conformation, and this binding mode seems to be essential for Wnt inhibition. NCB-0846 suppresses Wnt-driven intestinal tumorigenesis in $Apc^{min/+}$ mice and the sphere- and tumour-forming activities of colorectal cancer cells. TNIK is required for the tumour-initiating function of colorectal cancer stem cells. Its inhibition is a promising therapeutic approach.

[1] Division of Chemotherapy and Clinical Research, National Cancer Center Research Institute, 5-1-1 Tsukiji, Chuo-ku, Tokyo 104-0045, Japan. [2] Carna Biosciences, Inc., BMA 3F 1-5-5 Minatojima-Minamimachi, Chuo-ku, Kobe 650-0047, Japan. [3] Division of Structural and Synthetic Biology, RIKEN Center for Life Science Technologies, 1-7-22 Suehiro-cho, Tsurumi-ku, Yokohama 230-0045, Japan. [4] Division of Cancer Differentiation, National Cancer Center Research Institute, 5-1-1 Tsukiji, Chuo-ku, Tokyo 104-0045, Japan. * These authors contributed equally to this work. Correspondence and requests for materials should be addressed to T.Y. (email: tyamada@ncc.go.jp).

The great majority of colorectal cancers carry somatic mutations in one of two genes involved in the canonical Wnt/β-catenin signalling pathway: the adenomatous polyposis coli (APC) and β-catenin (CTNNB1) genes[1–4]. The high frequency of Wnt signalling gene alterations (>90%) was reconfirmed by a recent large-scale sequencing project of TCGA (The Cancer Genome Atlas)[5]. Wnt signalling is essential for maintaining the undifferentiated status and self-renewal capability of intestinal stem cells[6], and constitutive activation of Wnt signalling due to such genetic alterations most likely gives rise to tumour-initiating or cancer stem cells (CSCs)[7–9]. In fact, two representative colorectal CSC markers, CD44 and leucine-rich repeat-containing G-protein-coupled receptor 5 (Lgr5), are the known target gene products of Wnt signalling[10–12]. A CSC develops into a microscopic adenoma, and eventually into invasive carcinoma, following sequential accumulation of other genetic alterations[1].

Loss of APC is the earliest genetic event during colorectal carcinogenesis[13], but established colorectal cancer cells still retain dependency on Wnt signalling[9,14]. Numerous attempts have therefore been made to develop therapeutics targeting the Wnt/β-catenin pathway. In colorectal cancer, however, due to the genetic inactivation of APC, only the molecules downstream of APC can be considered as therapeutic targets[15]. TNIK was identified independently by two research groups[16,17] through comprehensive mass-spectrometry analyses of proteins associated with the T-cell factor-4 (TCF4) and β-catenin transcriptional complex[16,18], the most downstream effector of Wnt signalling. TNIK was essential for full activation of Wnt signalling, and colorectal cancer cells were highly dependent on TNIK for growth[17]. On the basis of these results, TNIK has been considered a promising drug target. Here we report the development of a novel small-molecule TNIK inhibitor having anti-Wnt and anti-cancer stem cell activities. TNIK has various functions other than regulation of Wnt signalling[19,20]. We therefore initiated this study by developing Tnik-deficient mice to confirm the safety and feasibility of TNIK targeting.

## Results

**Tnik-deficient mice show reduced tumorigenicity**. A targeting construct was designed to delete exon 4 of the murine Tnik gene (Fig. 1a). It was anticipated that this deletion would remove almost the entire kinase domain (amino-acid residues 25–289) including the ATP-binding site. Consistent with genotyping (Fig. 1b, left) homozygous Tnik mutant (C57BL/6J-$Tnik^{-/-}$) mice were confirmed to lack expression of the targeted exon (Fig. 1b, right). The Tnik-knockout mice were born alive and apparently developed normally for up to 2 years after birth. No body weight reduction has been reported in Tnik-knockout mice with targeted deletion of exons 6 and 7 (ref. 21), but the body weight of mice lacking exon 4 of Tnik (this study) was significantly lower than that of wild-type littermates (Supplementary Fig. 1). Among the STE20 (Sterile 20) family genes (Supplementary Fig. 2a), two close relatives of Tnik—Mink and Nrk— were compensatorily upregulated (Supplementary Fig. 2b). TCF4/ TCF7L2 (transcription factor-7 like 2) has been implicated in glucose and lipid metabolism, and its gene polymorphism has been linked to the development of diabetes[22,23]. The serum levels of total cholesterol (T-CHO), glucose (GLU) and high-density lipoprotein cholesterol (HDL-C) were significantly decreased in $Tnik^-$-deficient mutant mice (Supplementary Fig. 3), but no apparent histological abnormality was observed in their major organs (Supplementary Fig. 4).

Treatment with Wnt3a induced transcriptional activity of TCF/LEF (lymphoid enhancer factor) in wild-type ($Tnik^{+/+}$) mouse-derived embryonic fibroblasts (MEFs), but the induction

was attenuated in heterozygous ($Tnik^{+/-}$) and homozygous ($Tnik^{-/-}$) mutant MEFs (Fig. 1c). The expression of Wnt target genes such as Myc and Cd44 was significantly reduced in $Tnik^{-/-}$ MEFs (Fig. 1d).

We then examined the sensitivity of Tnik-deficient mice to a potent colon carcinogen, azoxymethane (AOM)[24] (Fig. 1e). AOM is known to initiate tumorigenesis through induction of mutations in the Ctnnb1 gene[25]. We found that $Tnik^{-/-}$ mutant mice developed significantly fewer colon tumours after administration of AOM than wild-type $Tnik^{+/+}$ littermates. The mean tumour size in $Tnik^{-/-}$ mice was apparently increased (Fig. 1e, right), but not to a statistically significant degree.

$Apc^{min/+}$ mice carry a truncational mutation of the Apc gene and spontaneously develop tumours in the small intestine, and much less frequently in the colon[26]. We next produced combined mutant mice by crossing Tnik-deficient mice to $Apc^{min/+}$ mice to reveal the involvement of Tnik in Wnt-driven tumorigenesis. The number of tumours that developed in the small intestine and colon of $Apc^{min/+}/Tnik^{-/-}$ mice was significantly lower than that in the $Apc^{min/+}/Tnik^{+/+}$ littermates (Fig. 1f–h).

**Identification of a novel TNIK inhibitor**. Having confirmed the safety and feasibility of targeting TNIK, we then screened an in-house kinase-focused compound library and identified a series of quinazoline analogues with high TNIK enzyme-inhibitory activity. Subsequent lead optimization resulted in the identification of NCB-0846 [cis-4-(2-(3H-benzo[d]imidazol-5-ylamino)-quinazolin-8-yloxy)cyclohexanol] (Fig. 2a, left). Structure-activity relationship analysis revealed that stereochemistry at the chiral terminal hydroxyl group of the cyclohexane moiety was important for inhibition of the TNIK enzyme. NCB-0846 showed inhibitory activity against TNIK with an half-maximal inhibitory concentration ($IC_{50}$) value of 21 nM (Fig. 2b), but its diastereomer (named NCB-0970) having the opposite configuration at the terminal hydroxyl group (Fig. 2a, right) showed 13-fold lower TNIK-inhibitory activity ($IC_{50} = 272$ nM) (Fig. 2b). NCB-0846 also inhibited FLT3, JAK3, PDGFRα, TRKA, CDK2/CycA2, and HGK (>80% at 0.1 μM; Supplementary Table 1). NCB-0970 showed a similar inhibitory profile over the human kinome except for STE20 member kinases including TNIK (Fig. 2c). On the basis of these results, we used NCB-0970 as a negative control for TNIK inhibition thereafter.

TCF4 is a substrate of the TNIK enzyme. TNIK phosphorylates TCF4 and regulates its transcriptional activity[17]. NCB-0846 induced faster migration of TCF4 phosphorylated by TNIK within a concentration range of 0.1–0.3 μM (*, Fig. 2d) and completely inhibited the phosphorylation of TCF4 at a concentration of 3 μM (Fig. 2d). Furthermore, NCB-0846 blocked the auto-phosphorylation of TNIK (Fig. 2e). NCB-0846 inhibited the TCF/LEF transcriptional activity of Wnt3a-treated HEK293 (Fig. 2f) and HCT116 (carrying CTNNB1 mutation) and DLD-1 (carrying APC mutation) colorectal cancer (Fig. 2g) cells. NCB-0846 reduced the expression of the Wnt target genes AXIN2 and MYC as well as that of TNIK, but the expression of CCND1 was not affected (Fig. 2h). This is in agreement with our previous data obtained using small-interfering RNA (siRNA) against TNIK[17]. NCB-0846 also reduced the expression of TNIK, AXIN2 and cMYC at the protein level (Fig. 2i). TNIK has been reported to regulate the stability of a Wnt co-receptor, low-density lipoprotein receptor-related protein 6 (LRP6)[27]. Consistent with this, LRP6 and LRP5 were down-regulated by NCB-0846 (Fig. 2i; Supplementary Fig. 5).

**NCB-0846 inhibits cancer cell growth in vitro and in vivo**. NCB-0846 showed 6.8-fold higher cell growth-inhibitory activity against HCT116 cells than NCB-0970 under conventional

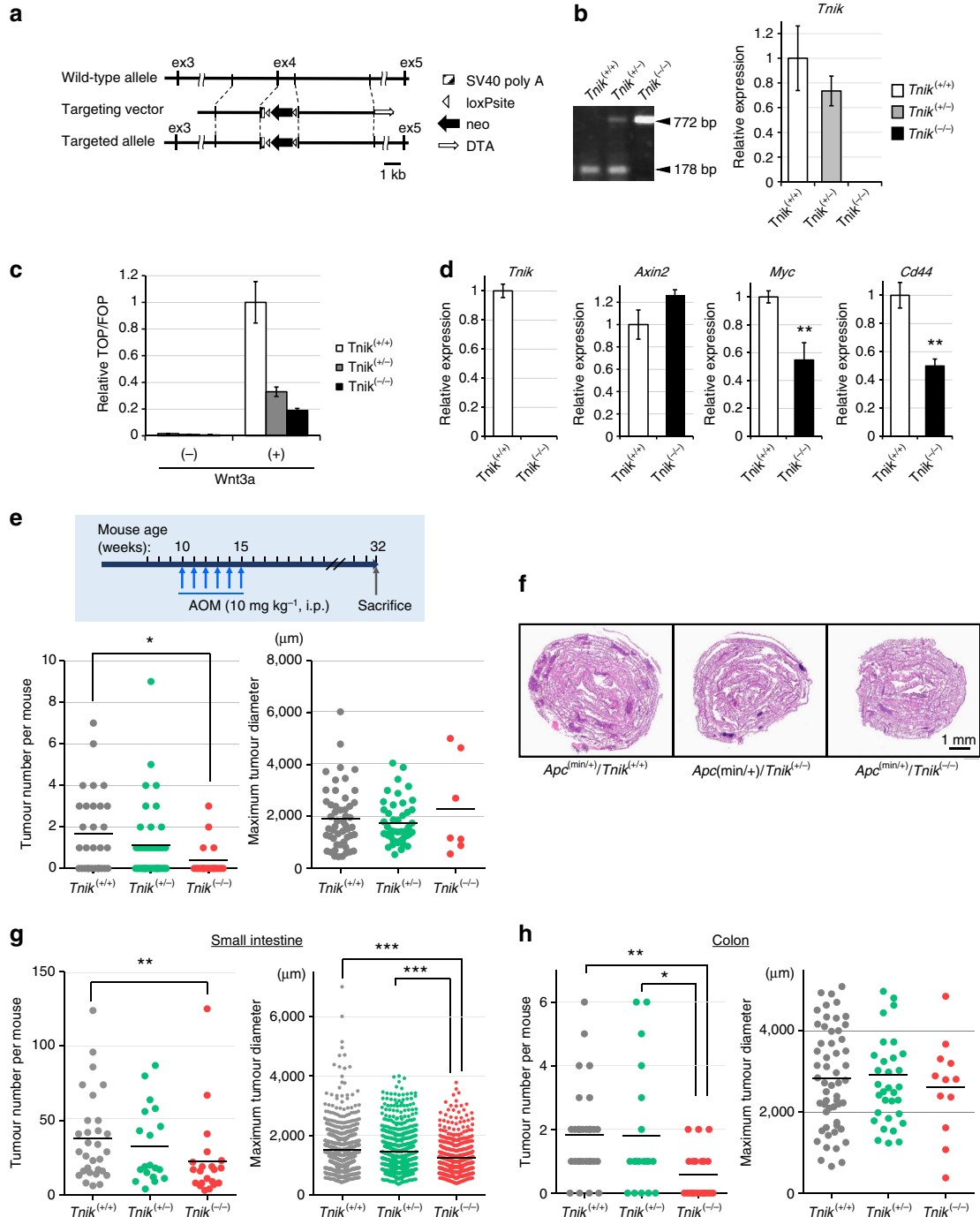

**Figure 1 | Reduced tumorigenicity of *Tnik*-deficient mice.** (**a**) Schematic representation of the *Tnik* allele and the targeting construct with the neomycin (neo) cassette driven by the phosphoglycerate kinase I (PGK) promotor as a positive selection marker and diphtheria toxin (DTA) as a negative selection marker. (**b**) Representative PCR-based genotyping (left) and relative *Tnik* gene expression (normalized to β-actin (*Actb1*)) in the brains of 10-week-old male wild-type (C57BL/6J-*Tnik*+/+), heterozygous mutant (*Tnik*+/−), and homozygous mutant (*Tnik*−/−) mice (right). Error bars represent s.e.m. (**c**) TCF/LEF transcriptional activity of MEFs derived from *Tnik*+/+, *Tnik*+/− and *Tnik*−/− embryos. MEFs were transfected with the superTOP-flash or superFOP-flash luciferase reporter in triplicate and treated with 30 ng ml−1 recombinant Wnt3a for 24 h or not. (**d**) Relative expression of *Tnik* and Wnt target genes (*Axin2*, *Myc* and *Cd44*) by MEFs derived from *Tnik*+/+ and *Tnik*−/− embryos. (**e**) Schedule of AOM treatment (top) and the mean number and maximum diameter of tumours developed in male *Tnik*+/+ (n = 30), *Tnik*+/− (n = 40) and *Tnik*−/− (n = 18) mice at the age of 32 weeks (bottom). *P < 0.05. (**f**) Macroscopic (so-called Swiss roll) view of whole small intestines of representative *Apc*min/+/*Tnik*+/+, *Apc*min/+/*Tnik*+/− and *Apc*min/+/*Tnik*−/− mice. (**g,h**) Mean numbers and maximum diameters of tumours that developed in the small intestine and colon of 16-week-old male *Apc*min/+/*Tnik*+/+ (n = 30), *Apc*min/+/*Tnik*+/− (n = 19) and *Apc*min/+/*Tnik*−/− (n = 21) mice. ***P < 0.001; **P < 0.01; *P < 0.05.

two-dimensional (2D) culture conditions (Fig. 3a). Meanwhile, compared with NCB-0970, NCB-0846 showed much higher (~20-fold) inhibitory activity against colony formation by the same cells in soft agar (Fig. 3b), indicating that this compound has more potent activity against the clonogenicity of cancer cells. NCB-0846 was administrable orally and suppressed the growth of

tumours established by inoculating HCT116 cells into immuno-deficient mice (Fig. 3c, left). The body weight of mice fell at the beginning of NCB-0846 administration, but gradually recovered (Fig. 3c, right). The expression of Wnt-target genes (*AXIN2*, *MYC* and *CCND1*) in xenografts was reduced following the administration of NCB-0846 (Fig. 3d). NCB-0846 induced an increase in the sub-G1 cell population (Fig. 3e). Cleavage of

poly (ADP-ribose) polymerase 1 (PARP-1; Fig. 3f) indicated the induction of apoptosis.

The effect of NCB-0846 on Wnt-driven tumorigenesis was then investigated in *Apc*[min/+] mice. The hydrochloride salt of NCB-0846 was water-soluble and used for oral administration subsequently. Administration of NCB-0846 was started 10 weeks after birth, and the mice were scarified at the age of 15 weeks.

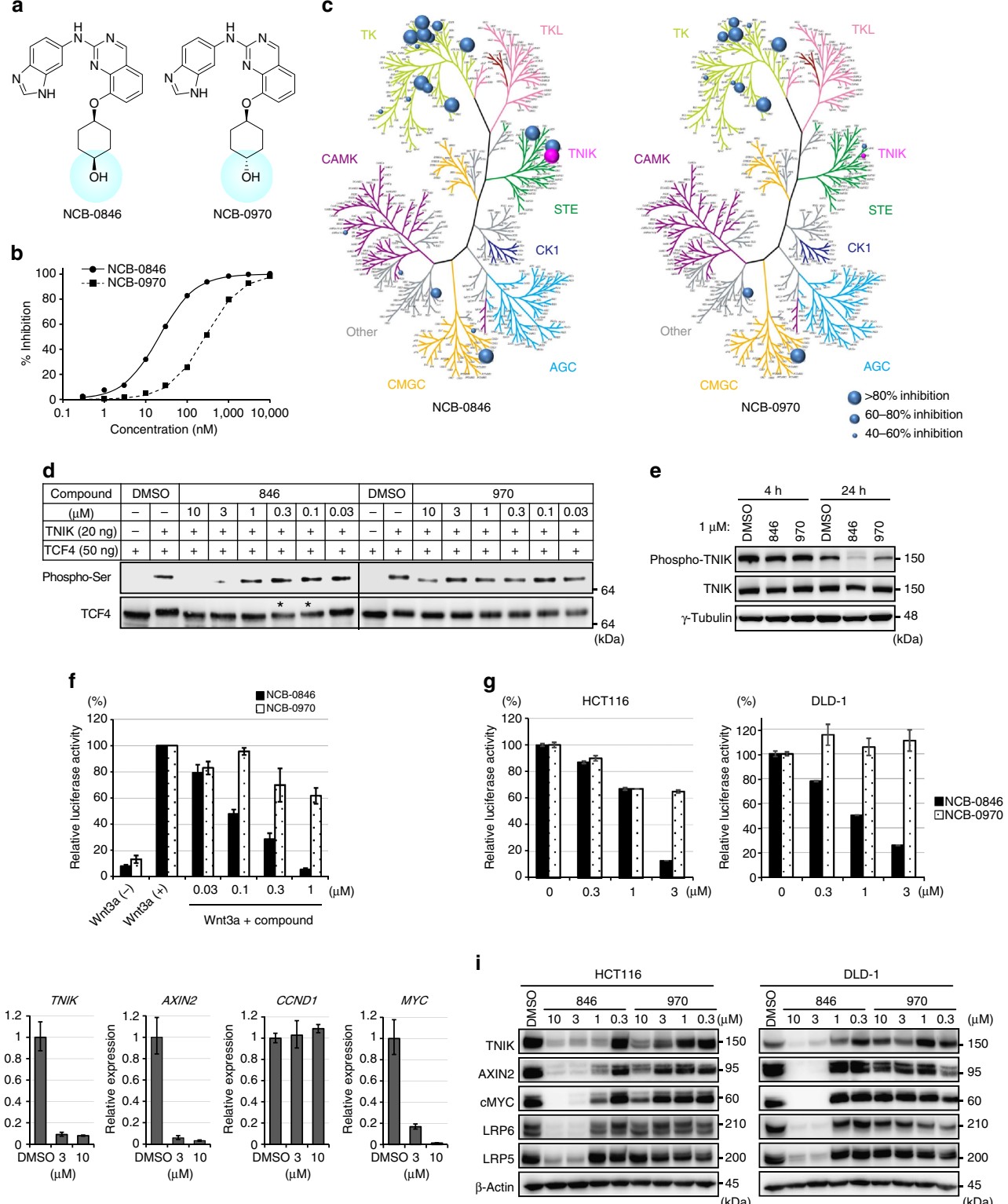

NCB-0846 dose dependently reduced the multiplicity and dimensions of tumours that developed in the small intestine (Fig. 3g). Although the effect of NCB-0846 on colon tumorigenesis was not statistically significant, this finding may not have been conclusive in view of the small number of animals examined and the general paucity of colon tumour development in the $Apc^{min/+}$ mice. Active Wnt signalling transforms a single intestinal epithelial cell into adenoma, and NCB-0846 is thought to suppress this tumour initiation process. The number of tumours that developed in the colon was also reduced, but the difference failed to reach statistical significance due to the paucity of colon tumours and the number of animals examined.

**Structural basis of TNIK inhibition**. Recently, a series of 4-phenyl-2-phenylaminopyridine-based small-molecule TNIK inhibitors have been reported by other investigators[28]. One of these compounds [4-methoxy-3-(2-(3-methoxy-4-morpholinophenylamino) pyridin-4-yl)benzonitrile] (named Compound 9) inhibited the enzyme activity of TNIK with an $IC_{50}$ value of 8 nM, but paradoxically had minimal Wnt signal-inhibitory effects ($IC_{50} > 2\,\mu$M)[28]. This paradox might be explainable by a difference in TNIK-binding modes between the reported compounds and ours. The crystal structure of the TNIK kinase domain (TNIK-KD) bound to either NCB-0846 or Compound 9 was determined at 2.9 or 2.4 Å resolution, respectively (Fig. 4; Table 1). For comparison, the apo structure of TNIK-KD was also determined at 2.9 Å resolution.

The apo (unbound) and NCB-0846-bound TNIK-KD structures were almost completely superposable (Fig. 4b). NCB-0846 binds to the ATP-binding site in a 16-membered ring cyclic conformation by forming an intramolecular hydrogen bond between the benzimidazole N3 and the cyclohexanol O4 (Fig. 4c; Supplementary Fig. 6). NCB-0846 interacts with the hinge region of TNIK-KD by forming two hydrogen bonds with the backbone carbonyl and amide groups of Cys108. In addition, the terminal hydroxyl group of the cyclohexanol forms another hydrogen bond with the main-chain carbonyl group of Gln157. This interaction with Gln157 is probably responsible for the potency of NCB-0846, because the diastereomer with the opposite configuration at this hydroxyl group (NCB-0970; Fig. 2a, right) showed impaired activity. Overall, the apo and NCB-0846-bound TNIK-KDs both adopt an inactive conformation. The RS2 (Phe172) and RS3 (Leu73) residues in the hydrophobic R-spine[29] are not correctly assembled, and the salt bridge between Lys54 of β3 and Glu69 of the αC-helix is disrupted in these structures (Fig. 4e).

In comparison with the apo and NCB-0846-bound TNIK-KD, the αC-helix of Compound 9-bound TNIK-KD is shifted to the 'in' conformation (Fig. 4b), similarly to the previously deposited model of TNIK with the Wee1Chk1 inhibitor (PDBID: 2X7F).

Compound 9 forms two hydrogen bonds with the backbone of Cys108, similarly to NCB-0846 (Fig. 4d; Supplementary Fig. 6). However, the nitrogen atom of the terminal nitrile group forms a water-mediated hydrogen bonding network including Glu69 and the backbone amide group of Asp171 in the DFG motif (Fig. 4e). This interaction stabilizes the DFG motif, allowing assembly of the RS2/RS3 residues and salt-bridge formation between Lys54 and Glu69. These structural features resemble the active conformation of protein kinase A (PKA, PDBID: 1ATP; Fig. 4f). Therefore, NCB-0846 and Compound 9 bind to the inactive and active conformations of TNIK-KD, respectively.

**TNIK inhibition abrogates colorectal cancer stemness**. Active Wnt signalling has been implicated in CSC function. NCB-0846, but not NCB-0970, downregulated the expression of putative colorectal CSC markers: CD44, CD133, and aldehyde dehydrogenase-1 (ALDH1)[30,31] (Fig. 5a, left). Flow cytometry analyses revealed that NCB-0846 reduced the proportion of cells showing high expression of CSC surface markers (CD44, CD133, CD166, CD29 and EpCAM) (Supplementary Fig. 7) and ALDH activity (Fig. 5b). CSCs often exhibit the epithelial–mesenchymal transition (EMT) phenotype[32]. NCB-0846 also reduced the expression of mesenchymal markers (Slug, Snail, Twist, Smad2 and Vimentin; Fig. 5a, right). However, embryonal stem cell markers (Oct4, Nanog and Sox2)[33] were not affected (Fig. 5a, right), suggesting that TNIK regulates the stemness of cells committed to intestinal epithelium.

Sphere formation in suspension culture is known to be the most prominent characteristic of CSCs and can be used for in vitro measurement of CSC function. Colorectal cancer cells have high sphere-forming activity because of their constitutive active Wnt signalling. Limiting dilution analysis (LDA)[34] revealed that short-term (3–4 days) treatment of colorectal cancer HCT116 and DLD-1 cells with NCB-0846 significantly abrogated their sphere formation activity (Fig. 5c and Supplementary Table 2).

CSCs are thought to have high tumorigenic activity, and a tumour can be formed even from a single CSC[35]. We found that inoculation of as little as 10 cells of bulk HCT116 or DLD-1 was sufficient to form a tumour in an immunodeficient mouse. However, short-term in vitro treatment with NCB-0846, but not with NCB-0970, significantly reduced tumour formation by the same numbers of HCT116 and DLD-1 cells (Fig. 5d; Supplementary Fig. 8).

**Patient-derived cancer-initiating cells**. Finally, the anti-tumour activity of NCB-0846 was examined in two more clinically relevant mouse models, in which xenografts were established from

**Figure 2 | Identification of a novel TNIK inhibitor.** (**a**) Structure of NCB-0846 (left) and its diastereomer NCB-0970 having the opposite configuration at the terminal hydroxyl group (right). (**b**) Inhibition of TNIK enzyme activity by the indicated concentration of NCB-0846 or NCB-0970. (**c**) Inhibitory profiles of NCB-0846 and NCB-0970 across the human kinome. The degree of inhibition is presented schematically by the size of the circles. (**d**) Inhibition of TCF4 phosphorylation by NCB-0846. The recombinant human TCF4 protein was incubated with the recombinant human TNIK protein in the presence of the indicated concentration of NCB-0846 or NCB-0970 and analysed by immunoblotting. (**e**) Inhibition of TNIK auto-phosphorylation by NCB-0846. HCT116 cells were cultured in the presence of DMSO (vehicle), 1 μM NCB-0846, or 1 μM NCB-0970 for 4 or 24 h and then analysed by immunoblotting with anti-phosphorylated TNIK, anti-TNIK, and anti-γ-tubulin (loading control) antibodies. (**f**) Suppression of TCF/LEF transcriptional activity by NCB-0846. HEK293 cells were transfected with pGL4.49[luc2P/TCF-LEF-RE/Hygro] and then treated with the indicated concentration of NCB-0846 or NCB-0970 for 24 h. The cells were then incubated with or without recombinant Wnt3a (10 ng ml$^{-1}$) for 5 h before measurement of luciferase activity. Data are represented as ratios relative to the DMSO control (set at 100%) ± s.e.m. (**g**) HCT116 (left) and DLD-1 (right) cells were transfected with one of the luciferase reporter plasmids (SuperTOP-flash or SuperFOP-flash) and then treated with 20 mM LiCl and the indicated concentration of NCB-0846 or NCB-0970 for 24 h. Data are presented as the ratios of SuperTOP-flash to SuperFOP-flash (TOP/FOP ratio). Error bars represent s.d. (**h**) HCT116 cells were treated with DMSO (control) or the indicated concentration of NCB-0846. Twenty-four hours later, the expression levels of the *TNIK*, *AXIN2*, *CCND1* and *MYC* genes (normalized to *ACTB*) were quantified in triplicate by RT–PCR. The values of cells treated with DMSO were set to one. Error bars represent s.d. (**i**) HCT116 and DLD-1 cells were treated with DMSO (vehicle) or the indicated concentration of NCB-0846 or NCB-0970 for 48 h. The expression of TNIK, AXIN2, c-MYC, LRP6, LRP5 and β-actin (loading control) proteins was analysed by immunoblotting.

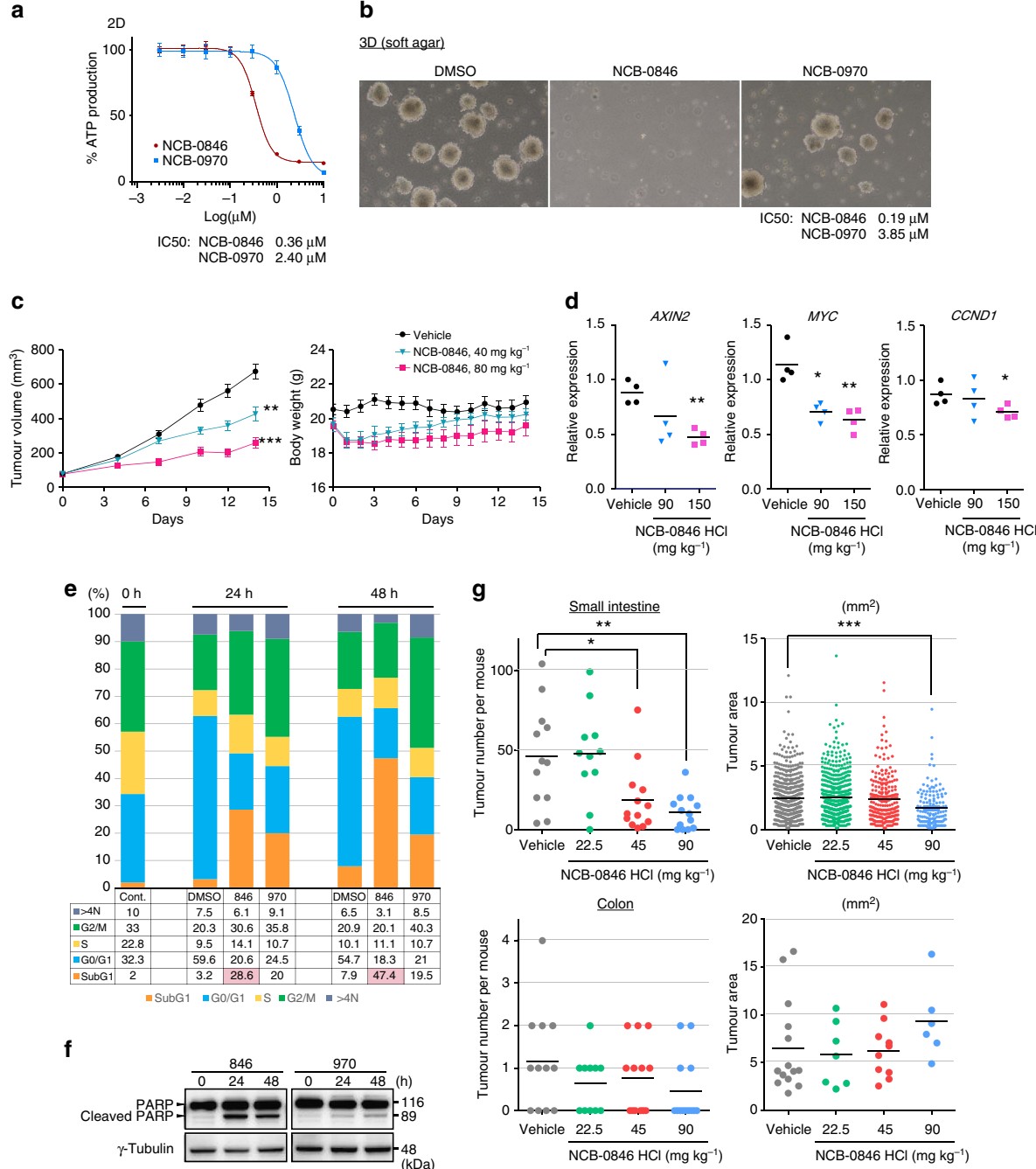

**Figure 3 | NCB-0846 inhibits cancer cell growth *in vitro* and *in vivo*.** (**a**) ATP production of HCT116 cells cultured with escalating doses of NCB-0846 or NCB-0970 for 72 h. $IC_{50}$ values were obtained by fitting a four-parameter dose–response curve to normalized data using GraphPad Prism software. Data represent the mean of six replicates. (**b**) Inhibition of colony formation by HCT116 cells in soft agar. Cells were embedded in top agar, covered by media containing DMSO, NCB-0846 (1 μM) or NCB-0970 (1 μM), and cultured for 14 days. (**c**) HCT116 cells were inoculated into the subcutaneous tissues of 9-week-old female BALB/c-nu/nu mice. When the volume of the xenografts reached ∼80 mm$^3$, the mice were administered daily with 0 (vehicle alone, $n = 9$), 40 ($n = 9$) or 80 mg kg$^{-1}$ ($n = 9$) BID (bis in die) NCB-0846. **$P < 0.01$, ***$P < 0.001$ (Dunnett's multiple comparison test). Error bars represent s.e.m. (**d**) Pharmacodynamics analysis. Mice carrying HCT116 xenografts of ∼650 mm$^3$ received a single oral dose of 0 (distilled water, vehicle control, $n = 4$), 90 ($n = 4$) or 150 mg kg$^{-1}$ ($n = 4$) NCB-0846 HCl (hydrochloride salt of NCB-0846). The tumours were excised, and the expression of *AXIN2*, *MYC* and *CCND1* was quantified by RT–PCR and normalized to *ACTB*. *$P < 0.05$, **$P < 0.01$ (Student's *t* test). (**e**) HCT116 cells were untreated (0 h) or treated with DMSO (control), NCB-0846 (3 μM) or NCB-0970 (3 μM) for 24 or 48 h. The percentage of cells in each cell cycle fraction was determined by flow cytometry. (**f**) HCT116 cells were treated with DMSO (control), NCB-0846 (1 μM) or NCB-0970 (1 μM) for 24 or 48 h. The expression of PARP-1 and γ-tubulin (loading control) was determined by immunoblotting. (**g**) Effects of NCB-0846 HCl on tumorigenicity in the small intestine and colon of *Apc*$^{min/+}$ mice. NCB-0846 HCl or water (VEH, vehicle) was orally administered BID for 35 days (10–15 weeks after birth). Data represent mean ± s.e.m of 11–13 mice (Vehicle, $n = 12$; 22.5 mg kg$^{-1}$, $n = 11$; 45 mg kg$^{-1}$, $n = 13$; 90 mg kg$^{-1}$, $n = 13$). ***$P < 0.001$, **$P < 0.01$ and *$P < 0.05$ (Dunnett's test) relative to the vehicle-treated group.

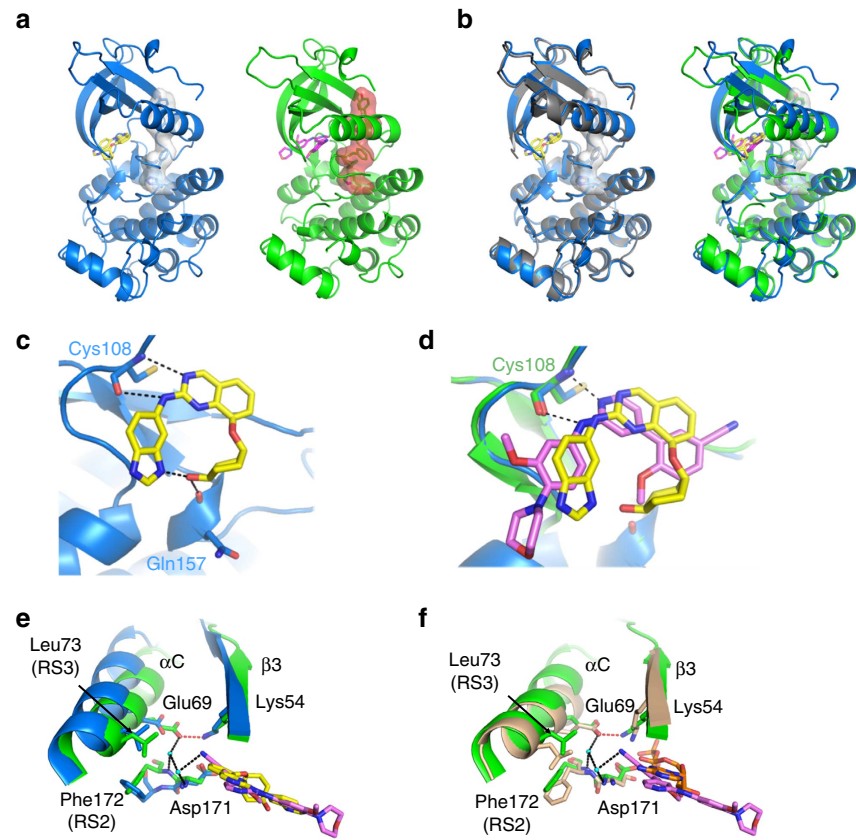

**Figure 4 | Structural basis of TNIK inhibition.** (**a**) The crystal structures of NCB-0846-bound and Compound 9-bound TNIK-KDs (blue and green). NCB-0846 (yellow) and Compound 9 (magenta) are shown by sticks. The hydrophobic R-spine residues are shown by sticks with surface representations (white and red). (**b**) Superimpositions of apo TNIK-KD (grey, left panel) or Compound 9-bound TNIK-KD (green, right panel) onto NCB-0846-bound TNIK-KD (blue). (**c**) NCB-0846 in the ATP-binding site of TNIK-KD. Hydrogen bonds are indicated by dashed lines. (**d**) Comparison of Compound 9 (magenta) and NCB-0846 (yellow) binding sites. (**e**) Differences in αC-helix position and DFG motif between NCB-0846-bound TNIK-KD (blue) and Compound 9-bound TNIK-KD (green). In Compound 9-bound TNIK-KD, the Lys54/Glu69 salt bridge (red dashed line, 2.7 Å) and the hydrogen bonding network (black dashed lines) mediated by water molecules (cyan balls) are indicated. (**f**) Superimpositions of Compound 9-bound TNIK-KD (green) onto the active state of PKA (wheat) bound to ATP (orange sticks).

colorectal cancer patients without being subjected to conventional 2D cell culture. It is known that adhesion to the bottom of a plastic culture dish irreversibly changes the gene expression profiles of cancer cells isolated from human primary tumours[36].

We have previously reported the long-term cultivation of cancer-initiating spheroids directly from primary colon tumours[37]. The spheroids derived from two patients (#6 and #19) expressed activated (stabilized) β-catenin and CSC markers (CD44, CD133 and ALDH1; Supplementary Fig. 9a,b) and maintained the function of CSCs to reconstitute the cancer tissue architecture. When implanted into mice, the patient-derived spheroids formed tumours with tubular structures and expressed cytokeratin 20 (CK20), a marker of intestinal epithelial cell differentiation[38] (Supplementary Fig. 9c). Although the spheroids were highly susceptible to cell dissociation, knockdown of TNIK by lentiviral transfer of small-hairpin RNA (shRNA) (Supplementary Fig. 10a) significantly reduced the reconstitution of spheroids from single cells (Supplementary Fig. 10b).

NCB-0846 suppressed the TCF/LEF transcriptional activity of a spheroid clone (see METHODS) in a dose-dependent manner (Fig. 6a–c). NCB-0846 reduced the growth of spheroids #6 and #19 *in vitro* (Fig. 6d) and the proportion of cells with high expression of CSC markers (CD44, CD133, CD166, CD24 and CD29; Fig. 6e) and ALDH activity (Supplementary Fig. 11). NCB-0846 suppressed spheroid reconstitution (Supplementary Fig. 12)

to a degree comparable with that of shRNA (Supplementary Fig. 10).

The oral administration of NCB-0846 suppressed the growth of tumours established by xenografting spheroids #6 and #19 into immunodeficient mice (Fig. 6f). Immunohistochemistry revealed suppression of CD44 expression and a modest decrease of Ki67 expression in xenografts resected after the treatment with NCB-0846 (Fig. 6g).

The second model involved patient-derived xenografts (PDXs) established from two patients with colorectal cancer (COX021 and COX026) (Supplementary Figs 13–14). The glandular histological architectures of the tumours from the patients were well preserved in their corresponding PDXs, indicating retention of the tissue reconstruction capabilities of the CSCs. Again, oral administration of NCB-0846 significantly suppressed the growth of the PDXs established from the two patients.

## Discussion

Several therapeutic strategies for targeting various molecular components of the Wnt signalling pathway, including porcupine (LGK974 (ref. 39)), frizzled receptors (OMP-18R5 (ref. 40) and OMP-54F28 (ref. 41)), tankyrases (XAV939 (ref. 42) and JW55 (ref. 43)), and cAMP response element binding protein (CREB)-binding protein (CBP) (ICG-001 (ref. 44) and PRI-724 (ref. 45)),

**Table 1 | Data collection and refinement statistics (molecular replacement).**

|  | Apo TNIK-KD | TNIK-KD/NCB-0846 | TNIK-KD/compound 9 |
|---|---|---|---|
| *Data collection* |  |  |  |
| X-ray source | BL26B2, SPring-8 | BL26B2, SPring-8 | BL32XU, SPring-8 |
| Wavelength (Å) | 1.0 | 1.0 | 1.0 |
| Space group | $P22_12_1$ | $C121$ | $P22_12_1$ |
| Cell dimensions |  |  |  |
| *a, b, c* (Å) | 50.2, 123.9, 158.5 | 212.6, 125.0, 49.9 | 47.2, 123.2, 157.7 |
| α, β, γ (°) | 90.0, 90.0, 90.0 | 90.0, 96.3, 90.0 | 90.0, 90.0, 90.0 |
| Resolution (Å) | 48.6–2.9 (3.1–2.9) | 46.9–2.9 (3.1–2.9) | 48.5–2.4 (2.5–2.4) |
| $R_{merge}$ | 0.098 (0.692) | 0.117 (0.878) | 0.131 (0.787) |
| $I/\sigma I$ | 18.2 (3.4) | 15.8 (2.7) | 12.2 (2.9) |
| Completeness (%) | 100.0 (100.0) | 100.0 (100.0) | 99.9 (100.0) |
| Redundancy | 7.3 (7.5) | 7.7 (7.8) | 7.2 (7.4) |
| *Refinement* |  |  |  |
| Resolution (Å) | 48.6–2.9 | 46.9–2.9 | 48.5–2.4 |
| No. reflections | 22,686 | 28,789 | 36,883 |
| $R_{work}/R_{free}$ | 0.209/0.281 | 0.184/0.239 | 0.205/0.271 |
| No. atoms |  |  |  |
| Protein | 6,728 | 6,915 | 6,711 |
| Ligand/$SO_4$ | − / − | 84/15 | 93/10 |
| Water | 2 | 7 | 107 |
| *B*-factors |  |  |  |
| Protein | 63.0 | 62.9 | 43.4 |
| Ligand/$SO_4$ | − / − | 54.3/79.8 | 35.1/64.8 |
| Water | 44.5 | 44.5 | 36.9 |
| r.m.s. deviations |  |  |  |
| Bond lengths (Å) | 0.004 | 0.004 | 0.006 |
| Bond angles (°) | 0.791 | 0.822 | 0.982 |
| Ramachandran Plot |  |  |  |
| Favoured (%) | 94.5 | 94.9 | 96.0 |
| Allowed (%) | 5.4 | 5.1 | 4.0 |
| Outliers (%) | 0.1 | 0.0 | 0.0 |
| PDB ID | 5CWZ | 5D7A | 5AX9 |

r.m.s., root mean square.
Highest resolution shell is shown in parenthesis.

have been developed (Supplementary Fig. 5). Some of these are being evaluated in early-phase clinical trials. At present, these anti-Wnt therapeutics appear to be clinically safe (except for some bone effects[41]), and no long-feared adverse effects in the gastrointestinal tract have yet been observed. TNIK regulates Wnt signalling in the most downstream part of the pathway, and its pharmacological inhibition has been expected to block the signal even in colorectal cancer cells with mutation in the *APC* or *CTNNB1* gene.

In this study, we obtained preclinical proof of concept (POC) for this therapeutic approach. We developed a novel small-molecule compound (named NCB-0846) with high inhibitory activity against TNIK. NCB-0846 blocked Wnt signalling (Fig. 2) and showed marked anti-tumour (Fig. 3) and anti-CSC (Fig. 5) activities. However, these activities were not observed for 4-phenyl-2-phenylaminopyridine-based TNIK inhibitors reported by Ho et al.[28]. We revealed a structural basis for this difference (Fig. 4). Human TNIK consists of serine–threonine kinase and scaffold domains[21,46]. The scaffold domain is involved in intra/intermolecular protein–protein interaction. TNIK interacts directly with TCF4 and β-catenin, and regulates Wnt signalling[17].

We speculate that inhibition of kinase activity via simple occupation of the ATP-binding pocket by a small chemical molecule might be insufficient to fully abrogate the functions of TNIK, considering the observed differences in the active/inactive conformations of the kinase domain.

The observed reduction of tumour multiplicity in *Tnik*-deficient mice (Fig. 1) indicates that TNIK is involved in the initiation of tumorigenesis. This is consistent with the so-called gatekeeper function of the *APC* tumour suppressor, which inhibits the initiation of (or entry into) carcinogenesis[1]. We obtained similar results with NCB-0846, but the latter was found to reduce the size of tumours further (Fig. 3g). The difference may be explained by a compensatory increase (>2.5-fold) of *Mink* expression in *Tnik*-deficient mice (Supplementary Fig. 2b). However, we must also consider the off-target effects of NCB-0846. Although we carefully included a diastereomer (NCB-0970) as a control in every experiment, the blocking of other signalling pathways cannot be completely excluded. NCB-0846 inhibited FLT3, JAK3, PDGFRα, TRKA, CDK2/CycA2 and HGK as well as TNIK (Supplementary Table 1). Activation of these signalling proteins is potentially oncogenic, and the anti-tumour activities of

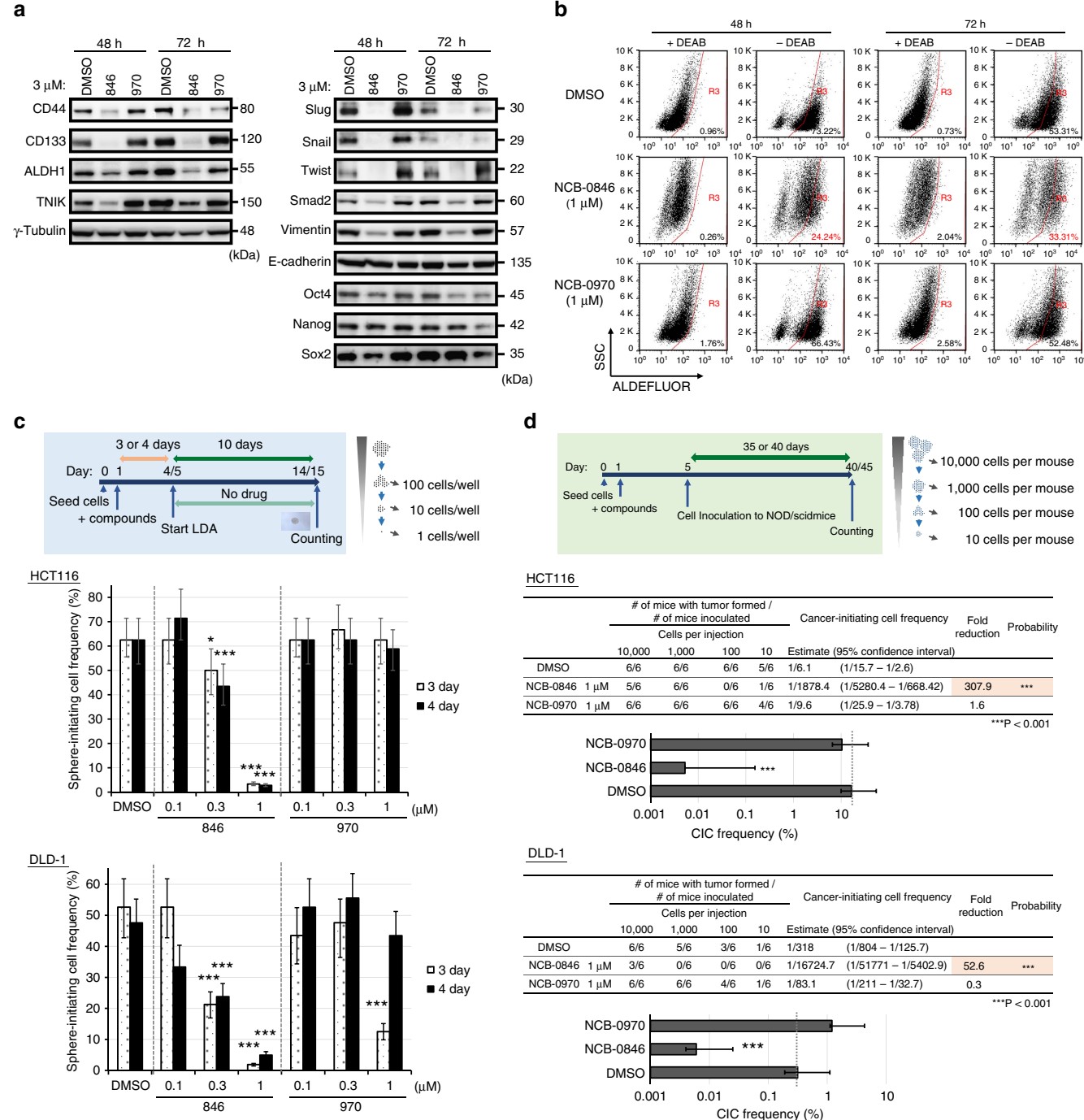

**Figure 5 | TNIK inhibition abrogates colorectal cancer stemness. (a)** HCT116 cells were treated with DMSO (vehicle), NCB-0846 (3 μM), or NCB-0970 (3 μM) for 48 or 72 h. The expression of the indicated proteins was analyzed by immunoblotting. **(b)** HCT116 cells were treated with DMSO (vehicle), NCB-0846 (1 μM) or NCB-0970 (1 μM) for 48 or 72 h. The population of cells with ALDEFLUOR intake in the presence (negative control) and absence of *N,N*-DEAB was determined by flow cytometry. Gates were set according to the highest intake of ALDEFLUOR by cells in the DEAB control. Abbreviation: SSC, side scatter. **(c)** Reduction in the frequency of sphere-forming cells by NCB-0846. HCT116 and DLD-1 cells were treated with DMSO, or with the indicated concentration of NCB-0846 or NCB-0970, for 3 or 4 days. Viable cells were distributed into 96-well U-bottomed culture clusters at a density of 1, 10 or 100 cells per well and cultured for 10 days. The frequency of sphere-forming cells was computed using ELDA software (***$P<0.001$ relative to DMSO). **(d)** Reduction of colorectal cancer cell tumour formation by NCB-0846. HCT116 and DLD-1 cells were treated with DMSO (control), NCB-0846 (1 μM) or NCB-0970 (1 μM) for 96 h. Viable cells were injected into the subcutaneous tissues of 6-week-old male NOD/ShiJic-scid mice at a density of 10, 100, 1,000 or $1 \times 10^4$ cells per mouse. Thirty-five (HCT116) and 40 (DLD-1) days later, the number of mice that had developed tumours was counted. The frequency of tumour-forming cells was calculated using ELDA software (***$P<0.001$ relative to DMSO). CIC, cancer-initiating cell.

NCB-0846 might be mediated not only by Wnt inhibition, but also by inhibition of some other pathways. It will be necessary to profile the entire signalling pathways affected by NCB-0846.

Development of a TNIK inhibitor with higher selectivity might be needed to address this issue. The structural information reported here (Fig. 4) would help identify such a compound.

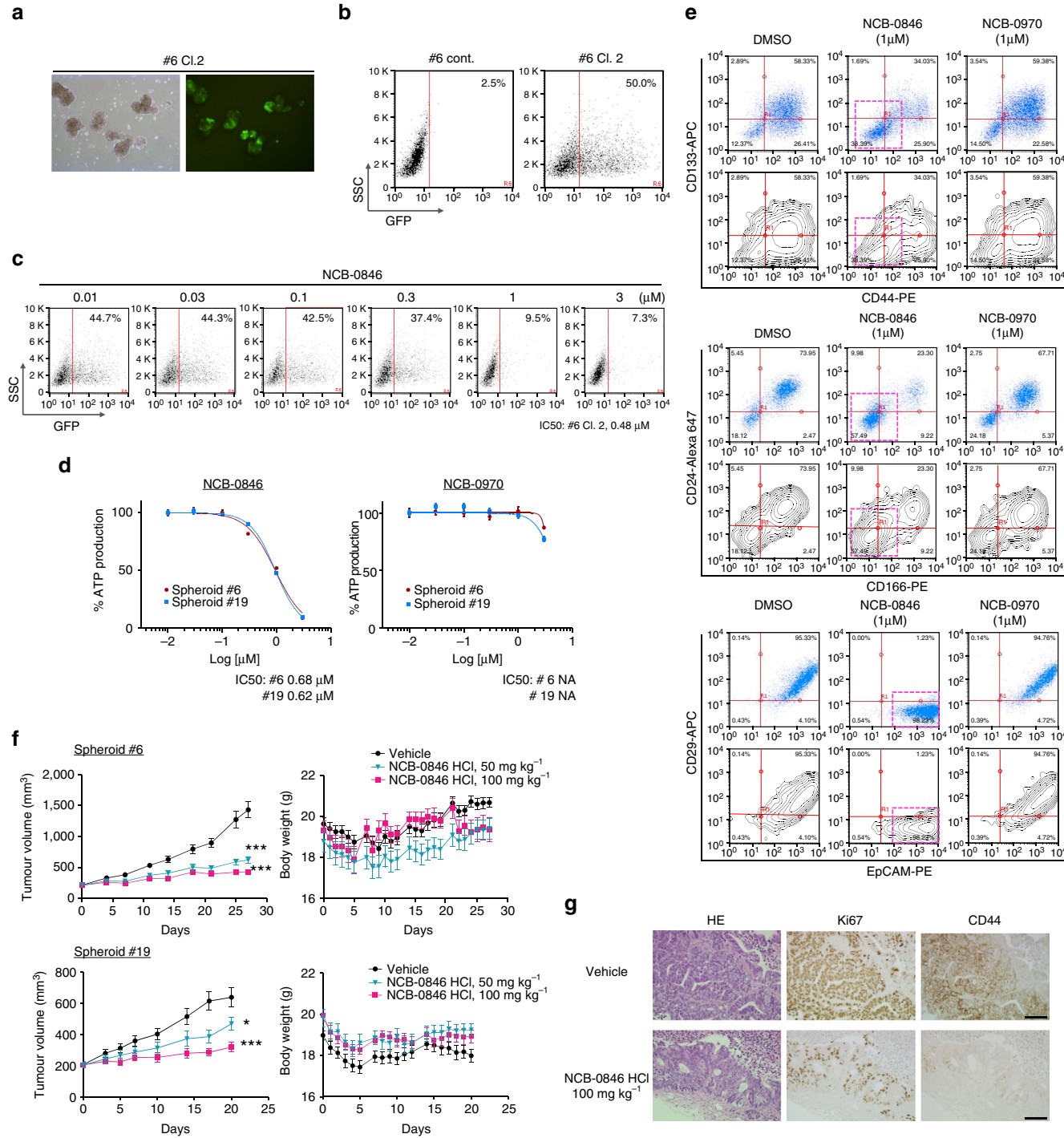

**Figure 6 | Patient-derived cancer-initiating cells.** (**a**) Phase-contrast (left) and fluorescence (right) microscopy of a spheroid clone (named #6 Cl. 2) expressing Wnt-responsive promoter-driven green fluorescent protein (GFP). (**b**) GFP expression of a spheroid clone #6 Cl. 2 (right) and a control clone (#6 Cont.) (left) determined by flow cytometry. (**c**) Dose-dependent reduction of cells expressing Wnt-driven GFP in a spheroid clone #6 Cl. 2 by NCB-0846. (**d**) ATP production of spheroids #6 and #19 cultured for 3 days in the presence of the indicated concentration of NCB-0846 or NCB-0970. NA, not available ( $>3\,\mu M$ ). (**e**) Spheroid #6 was treated with DMSO (vehicle), NCB-0846 ($1\,\mu M$), or NCB-0970 ($1\,\mu M$) for 96 h. The population of cells expressing CD133, CD44, CD24, CD166, CD29 and EpCAM was determined by dual-color flow cytometry. APC, allophycocyanin; PE, phycoerythrin. (**f**) NOD/ShiJic-scid mice engrafted with patient-derived colon cancer spheroid #6 ($1\times10^4$ cells) (top) or #19 ($1\times10^5$ cells) (bottom) were orally administered NCB-0846 HCl (0, 50, or 100 mg kg$^{-1}$, BID). *$P<0.05$, ***$P<0.001$ (Dunnett's test). Error bars represent s.e.m. (**g**) Excised tumours from mice engrafted with spheroid #6 and treated as described in **f** were analysed immunohistochemically with anti-Ki67 and anit-CD44v9 antibodies. Scale bar, 100 μm.

We conclude that TNIK is required for the tumour-initiating function of colorectal CSCs. However, TNIK is a multifunctional protein[19,20], and its function is not limited to regulation of Wnt signalling or colorectal carcinogenesis. Recently, amplification of *TNIK* has been reported in 7% (8/106) of gastric cancer patients[47]. Gastric cancer cells with such gene amplification

showed higher sensitivity to one of our earlier compounds (NCB-0001) than those without amplification. CSC often shows the EMT phenotype, and NCB-0846 inhibited the expression of mesenchymal marker proteins (Fig. 5a, right). Another earlier compound (NCB-0005 or KY-05009) was shown to inhibit EMT of lung cancer cells induced by transforming growth factor (TGF)-β1 (ref. 48). EMT has been implicated in cancer metastasis[49]. Conformational TNIK inhibitors may have a wide variety of clinical applications.

## Methods

**Ethical issues.** All the animal experimental protocols in this study were reviewed and approved by the institutional ethics and recombination safety committees of the National Cancer Center Research Institute (Tokyo, Japan). The minimum number of animals necessary for obtaining reliable results was used, and maximum attention was paid to animal rights and welfare protection.

**Generation of knockout mice.** A targeting vector (Fig. 1a) was constructed from the RPCI-23(RP23)-430J11 genomic DNA BAC (bacterial artificial chromosome) clone. The targeting vector was linearized by digestion and electroporated into mouse embryonic stem (ES) cells. Homologous recombinants were selected using G418 (Geneticin, Thermo Fisher Scientific) and injected into C57BL/6 mouse blastocysts. Chimeric male mice were crossed with C57BL/6J female mice to establish a heterozygous mutant line. C57BL/6J-$Apc^{min/+}$ mice[26] were purchased from the Jackson Laboratory.

**Genotyping.** Genotyping was performed by PCR using a pair of primers named Tnik null KO-F2 and Tnik null KO-R171 (generating a 178-bp fragment) for detecting the wild-type allele, and a pair of primers named Tnik null KO-F2 and NeoG02 (generating a 772-bp fragment) for detecting the targeted allele (Fig. 1b). $Apc^{min/+}$ mice were genotyped according to the supplier's instructions. The primer sequences are available on request.

**Real-time RT–PCR.** Total RNA was prepared with an RNeasy Mini Kit (Qiagen). cDNA was synthesized using SuperScript II reverse transcriptase (Thermo Fisher Scientific) or a High-Capacity cDNA reverse transcription kit (Thermo Fisher Scientific) and subjected to TaqMan gene expression assay using pre-designed primer and probe sets (listed in Supplementary Table 3). Amplification data measured as an increase in reporter fluorescence were collected using the StepOne Real-Time PCR System (Thermo Fisher Scientific). The relative mRNA expression level normalized to the internal control (human/mouse β-actin ($ACTB/Actb1$) gene) was calculated by the comparative threshold cycle ($C_T$) method[50]. Experiments were performed in triplicate and repeated at least two times.

**Blood chemical analysis.** Serum were collected from 10-week-old male $Tnik^{+/+}$ ($n=3$), $Tnik^{+/-}$ ($n=3$) and $Tnik^{-/-}$ ($n=3$) littermates. Biochemical analyses were performed at the Nagahama Bio-Laboratories Inc. (Supplementary Fig. 3).

**MEFs.** MEFs were collected from 13.5-day pregnant mice. Embryos were diced and trypsinized (Thermo Fisher Scientific) for 10 min at 37 °C (ref. 51). MEFs were cultivated in DMEM (Thermo Fisher Scientific) supplemented with 10% fetal bovine serum (Thermo Fisher Scientific), 100 U ml$^{-1}$ penicillin/streptomycin (Thermo Fisher Scientific), and 55 μM β-mercaptoethanol (Sigma-Aldrich; Fig. 1c,d).

**AOM treatment.** Ten-week-old mice were administered 10 mg kg$^{-1}$ body weight of AOM (Nard) by intraperitoneal injection once a week for 6 weeks, as described previously[24] (Fig. 1e).

**Evaluation of tumour multiplicity.** The mouse gut was filled with 10% buffered formalin (Sigma-Aldrich) via the anus immediately after sacrifice and opened longitudinally. The number and maximum diameter/area of tumours were measured by stereoscopic microscopy as described previously[26]. Differences at $P$ values of <0.05 were considered to be statistically significant.

**Mobility shift assay.** Enzymatic activity of TNIK was measured by mobility shift assay[52] using a QuickScout Screening Assist Kit (07-138MS, Carna Biosciences). The reaction product was quantified using a LabChip EZ Reader II (PerkinElmer). The IC$_{50}$ values were calculated from the dose–response curves using nonlinear regression analysis (Fig. 2b).

**Kinase selectivity profiling.** The selectivity of compounds against a panel of 50 human protein kinases was assessed using a non-radiometric assay[53]. Percentage

inhibition was determined at an inhibitor concentration of 0.1 M with ATP at the $K_m$ concentration (Fig. 2c and Supplementary Table 1).

**Phosphorylation of TCF4.** Recombinant human TCF4 (TCF7L2) protein tagged with glutathione S-transferase (GST) was obtained from Abnova (#H00006934-0P01). Fifty nanograms of TCF4 and 20 ng of the kinase domain of TNIK (1–314 amino-acid residues tagged with GST) were mixed in 40 μl of kinase buffer (20 mM HEPES (pH 7.5), 20 mM MgCl$_2$, 12.5 mM glycerophosphate, 0.01% Triton X-100 and 2 mM DTT) containing 50 μM ATP and incubated at room temperature for 1 h. The reaction was stopped by adding sample buffer (NuPAGE LDS Sample Buffer, Thermo Fisher Scientific) and analyzed by immunoblotting with anti-phosphoserine and anti-TCF4 antibodies (Fig. 2d).

**Cell lines.** The human embryonic kidney cell line HEK293 and the human colorectal cancer cell line DLD-1 were obtained from the Health Science Research Resources Bank (Osaka, Japan) and maintained in DMEM (Thermo Fisher Scientific) and RPMI (Thermo Fisher Scientific) supplemented with 10% fetal calf serum (Thermo Fisher Scientific), respectively. The human colorectal cancer cell line HCT116 was purchased from the American Type Culture Collection and cultured in RPMI supplemented with 10% fetal calf serum. These cell lines are not listed in the International Cell Line Authentication Committee database of cross-contaminated or misidentified cell lines. Absence of mycoplasma contamination was routinely confirmed using the e-Myco VALiD Mycoplasma PCR Detection Kit (iNtRON Biotechnology). All cell lines were authenticated by short tandem repeat profiling.

**Luciferase reporter assay.** A pair of luciferase reporter constructs, SuperTOP-flash and SuperFOP-flash (Addgene), were used to evaluate TCF/LEF transcriptional activity (Figs 1c and 2g). Cells were transiently transfected using lipofectamine 2000 (Thermo Fisher Scientific) in triplicate with one of the luciferase reporters and phRL-TK (Promega). Luciferase activity was measured with the Dual-luciferase Reporter Assay system (Promega) and normalized using *Renilla reniformis* luciferase activity as an internal control[54]. Assays were performed in triplicate and repeated at least two times.

HEK293 cells were transiently transfected with pGL4.49[luc2P/TCF-LEF-RE/Hygro] reporter vector (Promega) using Fugene HD Transfection Reagent (Promega; Fig. 2f). After overnight incubation, the cells were then treated with 10 mM LiCl (Sigma-Aldrich). Luciferase activity was measured using the ONE-Glo Luciferase Assay System (Promega). The assay was performed in triplicate and repeated at least two times.

In some experiments cells were treated with mouse Wnt3a (PeproTech) (Fig. 1c) or human Wnt3a (R&D Systems; Fig. 2f) before the measurements, as indicated in the figure legends.

**Chemical synthesis.** Details of the synthesis of compounds are available in the Supplementary Information.

**Antibodies.** Antibodies used in this study and their suppliers are listed in Supplementary Table 4. A rabbit monoclonal antibody against TNIK phosphorylated at the serine 764 residue (p-TNIK S764; Fig. 2e) was generated by immunizing rabbits with a synthetic peptide (CSSERTRVRAN(pS)KSEGSPVLPH) conjugated with keyhole limpet hemocyanin (KLH). After selection of positive hybridoma clones, a rabbit monoclonal antibody against p-TNIK S764 was purified from the culture medium of the hybridoma.

**Immunoblot analysis.** Protein samples were fractionated by SDS–PAGE and blotted onto Immobilon-P membranes (Millipore) as described previously[55]. After incubation with the primary antibodies (listed in Supplementary Table 4) at 4 °C overnight, the blots were detected with the relevant horseradish peroxidase-conjugated anti-mouse or anti-rabbit IgG antibody (Cell Signaling Technology) and Western lighting ECL Pro (PerkinElmer). Signals were visualized with the LAS-4010 system (GE Healthcare). Uncropped versions of the blots in Figs 2e and 3f are shown in Supplementary Fig. 15.

**Imunohistochemistry.** Immunoperoxidase staining (Fig. 6g) was performed using the Ventana DABMap detection kit and an automated slide stainer (Discovery XT, Ventana Medical Systems)[56].

For double immunofluoresence staining[57], antigen retrieval was performed in citrate buffer (pH 6.0) at 100 °C for 20 min. Tissue sections were incubated with primary antibodies (listed in Supplementary Table 4) diluted with Antibody Diluent (Dako) at 4 °C overnight, incubated with relevant secondary antibodies conjugated with Alxa Fluor 488 or Alxa Fluor 568, and mounted with Mount Permafluor (Thermo Fisher Scientific; Supplementary Fig. 9c).

**Drug sensitivity assay.** Cells were seeded at a density of 3,000 cells per well in 96-well plates. Twenty-four hours after seeding, the cells were exposed to serially

diluted compounds (0.003, 0.01, 0.03, 0.1, 0.3, 1, 3 and 10 µM) and incubated for 72 h (Fig. 3a). Patient-derived colorectal cancer spheroids were dissociated into single cells with Accutase (Nacalai Tesque) and seeded at a density of 3,000 cells per well into Ultra-Low attachment multiwall 96-well plates (Corning; Fig. 6d). ATP was measured with a CellTiter-Glo Luminescent Cell Viability Assay kit (Promega). Experiments were performed in triplicate and repeated at least two times.

**Soft-agar colony-formation assay.** One milliliter of 0.33% agar (Lonza) in culture medium containing $1.5 \times 10^4$ cells was plated onto 1 ml of solidified base agar (0.5%) in each well of six-well clusters. The top agar layer was covered with culture medium containing serially diluted compounds. The medium was replaced every three days. Cell number was determined colorimetrically at 450–650 nm using the Cell Count Reagent SF (Nacalai Tesque). $IC_{50}$ values were obtained by fitting a four-parameter dose–response curve to normalized data using GraphPad Prism version 5 (GraphPad software; Fig. 3b). Experiments were performed in triplicate and repeated at least two times.

**Cell cycle analysis.** Cells were dissociated with Accutase, fixed with 70% EtOH at − 20 °C, stained with Guava Cell Cycle reagent (Merck-Millipore) in accordance with the manufacturer's instructions, and analysed using a Guava easy Cyte HT flow cytometer (Merck-Millipore). Cell doublets were eliminated by doublet discrimination gating. Data were analyzed using FLOWJO version 10 software (Treestar; Fig. 3e). Experiments were performed in triplicate and repeated at least two times.

**Crystallization.** The DNA fragment of the kinase domain (residues 11–314) of human TNIK (TNIK-KD) was amplified by PCR and subcloned into the baculo-virus transfer vector pFastBacHT (Thermo Fisher Scientific) with an N-terminal $His_6$ tag and a thrombin cleavage site. The $His_6$-tagged TNIK-KD was expressed in Sf9 cells using a BAC-to-BAC Baculovirus Expression System (Thermo Fisher Scientific).

The infected Sf9 cells were lysed and sonicated in 20 mM Tris (pH 8.0), 1 M NaCl, 20 mM imidazole and 10% glycerol. The protein solution was applied to a HisTap column (GE Healthcare) and then eluted with a buffer containing 500 mM imidazole. The $His_6$ tag of the eluted sample was cleaved by thrombin (GE Healthcare) at 4 °C overnight. The solution was passed through a HisTap column again to remove the cleaved $His_6$ tag. The flow-through fraction was further purified on an ion-exchange column (MonoS, GE Healthcare) and a size-exclusion chromatography column (Superdex75, GE Healthcare) in a final buffer containing 20 mM HEPES (pH 7.0), 50 mM NaCl, 10 mM $MgCl_2$, 2 mM DTT and 10% glycerol. The sample was concentrated to 20 mg ml$^{-1}$ and stored at − 80 °C until use.

The apo TNIK-KD (20 mg ml$^{-1}$) was crystallized against a reservoir solution containing 0.2 M ammonium sulfate, 0.1 M bis–tris (pH 6.2), 35% PEG3350, and 9% ethylene glycol by the sitting drop vapor-diffusion method at 20 °C. For crystallization of the inhibitor-bound TNIK-KDs, the purified TNIK-KD protein (10 mg ml$^{-1}$) was mixed with a final concentration of 0.5 mM NCB-0846 or Compound 9 and 1% dimethyl sulfoxide (DMSO) and incubated at 4 °C for 1 h before crystallization. The reservoir solution contained 0.2 M ammonium sulfate, 0.1 M bis–tris (pH 5.6), 15% PEG3350 and 7% ethylene glycol for the NCB-0846-bound TNIK-KD and 0.2 M ammonium sulfate, 0.1 M bis–tris (pH 5.6), and 24% PEG3350 for the compound 9-bound TNIK-KD. The crystals of the apo TNIK-KD and the NCB-0846-bound TNIK-KD were directly flash-frozen in liquid nitrogen, while for the compound 9-bound TNIK-KD, 25% ethylene glycol was used as a cryoprotectant.

**X-ray data collection, structure determination, and refinement.** The diffraction data were collected at a wavelength of 1.0 Å at beamlines using the BL26B2 (the apo TNIK-KD and the NCB-0846-bound TNIK-KD) and BL32XU (the Compound 9-bound TNIK-KD), SPring-8 (Hyogo, Japan), and processed using the program XDS[58] and the CCP4 suite[59]. The structures were determined using the molecular replacement method with the program Phaser[60], using the coordinates of the TNIK-KD with wee1/chk1 inhibitor (PDB code: 2X7F) as the search model. The model was corrected with *COOT*[61] and refinement was performed with *PHENIX*[62]. Topology and parameter files for the inhibitors were generated using eLBOW of *PHENIX*. The Ramachandran statistics were analysed with MolProbity[63]. The graphical figures were drawn by *PyMOL* (http://www.pymol.org; Fig. 4).

**Cell surface marker analysis.** Cells were collected with Accutase, washed with stain buffer (BD Pharmingen), resuspended in 100 µl stain buffer ($5 \times 10^5$ cells per 100 µl), and incubated with fluorophore-conjugated antibodies or their corresponding isotype-matched control antibodies (Supplementary Table 4) at 4 °C for 30 min. The cells were then washed twice and resuspended in 500 µl of stain buffer. 7-Aminoactinomycin D (7AAD; BD Pharmingen) was added to the samples before analysis.

Relative fluorescence intensities were measured using a Guava easy Cyte HT flow cytometer following exclusion of dead cells on the basis of 7AAD

incorporation. Data analysis was performed using Guava InCyte software (Merck-Millipore). Experiments were performed in triplicate and repeated at least two times.

**ALDH activity.** Cells with ALDH activity were detected using an Aldefluor kit (Stem Cell Technologies) in accordance with the manufacturer's instructions. Briefly, cells were incubated in Aldefluor assay buffer containing ALDH substrate for 30 min at 37 °C. As a negative control, an aliquot of each sample was treated with 50 mM diethylaminobenzaldehyde (DEAB) to block ALDH activity.

Cells were analysed with a Guava easy Cyte HT flow cytometer. Data analysis was performed using the Guava InCyte software. The analysis was performed in triplicate and repeated at least two times.

**LDA.** One million HCT116 and DLD-1 cells were seeded onto 60-mm plates and treated with vehicle (DMSO) or compounds (0.1, 0.3 or 1.0 µM) for either 3 or 4 days. After washing off the compounds, the cells were dissociated to single cells in serum-free DMEM/F12 medium (Thermo Fisher Scientific) containing B27 (Thermo Fisher Scientific), 20 ng ml$^{-1}$ epidermal growth factor (EGF; Thermo Fisher Scientific), 10 ng ml$^{-1}$ basic fibroblast factor (bFGF; Thermo Fisher Scientific), 5 µg ml$^{-1}$ insulin (Thermo Fisher Scientific), 0.4% bovine serum albumin (Sigma-Aldrich) and 2 mM L-glutamine (Thermo Fisher Scientific). One hundred, 10 or 1 viable (determined by trypan-blue dye exclusion) cells were seeded into each well of a 96-well U-bottom PrimeSurface plate (Sumitomo Bakelite). Ten days later, the number of wells showing sphere formation was counted. The frequency of sphere-forming cells was calculated using ELDA software (http://bioinf.wehi.e-du.au/software/elda/index.html) provided by the Walter and Eliza Hall Institute (Fig. 5c; Supplementary Table 2). Experiments were performed in triplicate and repeated at least two times.

For evaluation of in vivo tumorigenicity (Fig. 5d and Supplementary Fig. 8), HCT116 and DLD-1 cells were treated with vehicle (DMSO), 1 µM NCB-0846 or 1 µM NCB-0970 for 4 days. After washing off the compounds, 10,000, 1,000, 100 or 10 viable cells in culture medium containing 50% Matrigel (BD Biosciences) were injected subcutaneously into the flanks of NOD/ShiJic-*scid* Jcl (6-week-old male) mice. The number of mice that developed tumours was counted, and the frequency of cells with tumour-forming capability was calculated using the ELDA software.

**Patient-derived spheroids.** Patient-derived cancer-initiating spheroids (#6 and #19)[37] were maintained in StemPro hESC SFM (Invitrogen) supplemented with 8 ng ml$^{-1}$ bFGF, 20 µM Y-27632 (Wako), and penicillin/streptomycin in ultra-low attachment culture dishes (Corning) or non-cell-adhesive PrimeSurface culture dishes (Sumitomo Bakelite) (Supplementary Fig. 9).

**Spheroid TCF/LEF reporter assay.** Lentiviral gene transfer of the reporter gene was used to evaluate the TCF/LEF transcriptional activity of patient-derived spheroids due to their poor transfection efficiency. Spheroids were infected with TCF/LEF reporter lentiviral particles encoding the green fluorescent protein (GFP) gene under the control of the TCF/LEF-responsive promoter (Cignal Lenti TCF/LEF Reporter (GFP) (Qiagen)) or relevant negative control lentiviral particles (Cignal Lenti negative control (GFP) (Qiagen)) at a multiplicity of infection of 10 in the presence of 4 µg ml$^{-1}$ SureEntry Transduction Reagent (Qiagen) for 24 h. GFP-positive spheroids were cloned by limiting dilution in the presence of 2 µg ml$^{-1}$ puromycin (Sigma-Aldrich; Fig. 6a,b).

The spheroid clones were seeded at a density of $1 \times 10^6$ cells per well in Ultra-Low attachment multiwall six-well plates (Corning) and cultured in the presence of serially diluted NCB-0846 (0.01, 0.03, 0.1, 0.3, 1 and 3 µM) for 7 days. GFP-positive cells were detected using the Guava easy Cyte HT flow cytometer (Fig. 6c). Experiments were performed in triplicate and repeated at least two times.

**LDA of spheroids.** Patient-derived spheroids were infected with lentiviral particles encoding shRNA for TNIK (Mission Lentiviral Transduction Particle, clone 34 (TRCN0000234734, Sigma-Aldrich)) or control lentiviral particles (MISSION Non-Target shRNA Control Transduction Partcles (SHC002V, Sigma-Aldrich)) at a multiplicity of infection of 5 in the presence of 8 µg ml$^{-1}$ hexadimethrine bromide (Sigma-Aldrich) for 24 h (Supplementary Fig. 10a).

Spheroids were gently dissociated into single cells with the use of StemPro Accutase (Thermo Fisher Scientific), and 1,000, 100, 10 or 1 viable cells were seeded into each well of a 96-well U-bottom PrimeSurface plate (Supplementary Figs 10b and 12). Experiments were performed in triplicate and repeated at least two times.

**Xenografts.** Five million HCT116 cells suspended in medium containing 25% Matrigel (BD Biosciences) were inoculated into the subcutaneous tissues of 9-week-old female BALB/c nude mice. When the tumour volume reached ∼80 mm$^3$, the mice were randomized according to tumour volume (9 mice per group). NCB-0846 suspended in DMSO/polyethylene glycol#400/30% 2-hydroxypropyl-β-cyclo-dextrin solution (10:45:45v/v) was administered daily by oral gavage at 0 (vehicle alone), 40 or 80 mg kg$^{-1}$ (body weight) BID (bis in die) for 14 days (Fig. 3c).

Patient-derived colorectal cancer spheroids were dissociated, resuspended in medium containing 50% Matrigel (Corning), and inoculated subcutaneously (#6; $1 \times 10^4$ cells per injection, #19; $1 \times 10^5$ cells per injection) into the flank of female NOD/ShiJic-scid mice (4-week-old females; CLEA Japan). When the tumour volume reached $\sim 200\,mm^3$, the mice were randomized according tumour volume (10 mice per group) and treatment was started. NCB-1026 (hydrochloride salt of NCB-0846) was dissolved in sterile saline solution and administered BID by oral gavage at 0 (vehicle alone), 50 or 100 mg kg$^{-1}$ (body weight) on a 6-days-on, 1-day-off schedule (Fig. 6f,g).

Patient-derived colorectal cancer xenografts COX021 and COX021 were established by Shanghai ChemPartner. Female Nu/Nu nude mice aged 5–6 weeks were implanted subcutaneously with the xenografts and randomized into three treatment groups (10 mice per group) based on the developed tumour volume ($\sim 200\,mm^3$). NCB-1026 (hydrochloride salt of NCB-0846) dissolved in sterile water was administered BID by oral gavage at 0 (vehicle alone), 45, or 90 mg kg$^{-1}$ (body weight) on a 5-days-on, 2-day-off schedule (Supplementary Figs 13 and 14).

Tumour volume was estimated based on the formula: (length × width$^2$)/2. The tumour volume (twice a week) and body weight (every day) of each mouse were recorded. The patient-derived spheroids and xenografts were established after obtaining informed consent from the donors.

**Data availability.** The data that support the findings of this study are available from the corresponding author upon request. Crystallographic data have been deposited in the Protein Data Bank (PDB) database with the accession numbers 5CWZ, 5D7A and 5AX9.

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

## Acknowledgements

We are grateful to Ms A. Sato, Dr T. Imai and Mr N. Uchiya (National Cancer Center Research Institute, Tokyo, Japan) for their help in the xenograft experiments, isolation of MEF, and preparation of paraffin-embedded tissue sections, respectively. This study was supported by the Program for Promotion of Fundamental Studies in Health Sciences conducted by the National Institute of Biomedical Innovation of Japan and by the National Cancer Center Research and Development Funds. The manuscript was reviewed by the Japan Agency for Medical Research and Development.

## Author contributions

M.Sa. and T.Y. designed the research; M.M., Y.U., H.O., A.M., N.O., H.M., S.K., T.I. and N.G. performed the experiments; M.K.-N., K.O., M.Sh., M.Sa. and T.Y. analysed the data and supervised the research; M.M., Y.U., A.M., N.O., M.Sa. and T.Y. prepared the manuscript.

## Additional information

**Competing financial interests:** Y.U., H.M., S.K, T.I., and M.Sa. are employees of Carna Biosciences Inc. (Kobe, Japan). The remaining authors declare no competing financial interests.

