## [Peer Review File · Nature Communications]

Reviewers' comments:

Reviewer #1 (Remarks to the Author):

Tesshi Yamada and colleagues present here an interesting paper, identifying a new orally available TNIK inhibitor, NCB-0846. The authors first show TNIK deficient mice to be resistant to Azoxymethane induced tumorigenesis. In a screen they identify NCB-0846 and its diastereomer NCB-0970. They test the efficacy of NCB-0846 in inhibiting cancer cell growth both in-vitro and in-vivo. Exploring the structural basis of TNIK inhibition, they discover that NCB-0846 binds to the inactive TNIK-KD which in turn confers Wnt signal -inhibitory effects. Finally using 2 mouse models, they evaluate the anti-tumor activity of NCB-0846.

The novelty of the paper lies in the identification of a new and potent Wnt signaling inhibitor NCB-0846, which could prove to be a promising therapeutic approach. However some of the data presented here are incomplete and need to be improved/repeated to validate the claims. While statistics have been used, quantification and details are needed in main figures and text.

Major revision :

- 1) The authors should provide more substantial data for the efficacy of the TNIK inhibitor in colon tumorigenesis. The data presented on Page 8 is insufficient and should either be repeated or details need to be added.
- 2) The authors do not consider the off -target effects of NCB-0846. For the identified TNIK inhibitor to be considered as a therapeutic, off target effects on other signaling pathways need to be ruled out. A gene expression profiling to identify these pathways might be beneficial. .
- 3) It is not appropriate to move the whole methods section to the supplementary.

Minor revision :

- 1) Page 6, Line 7 - Quantification is required. 'Small number of incidentally developed large tumors' is vague. Numbers need to be provided and added to text and figures.
- 2) Figure 1h- Can the authors comment on why the tumors increase in size in the colon? Quantification is required (how many were screened and how many had large tumors).
- 3) Page 8, Line 22 / Figure 3g- Data on colon is not satisfactory and needs to be improved.
- 4) Supplementary S4 could be improved.
- 5) While mouse models provide useful insights, the intestinal organoid technology provides a robust system for such drug efficacy studies. We suggest that the authors also test the potency of NCB-0846 on human tumor organoids (colon and small intestine).

Reviewer #2 (Remarks to the Author):

The manuscript written by Yamada and colleagues describes an extensive and thorough programme of research on the development and characterization of ATP-competitive TNIK kinase inhibitors. In my opinion, the study is well presented and interesting. There are aspects of it that could be approved, and I would ask the authors to respond to the following points:

1. Figure 1b -this data is not described clearly in the text - the sentence referring to this panel

describes the weight of mice, but does not mention the PCR results or the expression levels of TNIK (which is what the data shows). Could the authors also briefly explain their strategy (why an exon 4 deletion)?

2. Figure 2 - Could the authors comment on the apparently poor translation of in vitro potency into one of their cell-based assay (Fig. 2d)? Based on the immunoblot of anti-phosphoserine, the cellular IC50 appears to be in low micromolar range, about 100-fold higher than in vitro potency. This doesn't fit with much of the data that follows in subsequent figures, and it's important to resolve exactly what the potency of the inhibitor is against the target in cells.

3. Figure 2e - Is TNIK autophosphorylated in cells (reference), and why is loss of phosphorylation not observed after 4 hours?

4. Page 9/10 - The conformation of TNIK in apo form or bound to NCB-0846 is described as inactive, despite having DFG-in and C-helix in positions. This is confusing, and probably inaccurate, because the structure figures show differences in the DFG and C-helix in the structure of TNIK (bound to Compound 9) that looks like a truly active conformation. I think it would help here to include a figure panel showing the positions of these structural features compared to a well-known kinase in its active conformation such as PKA. Moreover, it would be useful to show the hydrophobic R-spine residues in these structures - this is a recent, and now widely accepted, concept and, based on the available images, it looks like there might well be a difference that would help to clarify whether the TNIK structures are indeed active or inactive. Then the authors we be able to write a more accurate description of the structures. For information on the R-spine, see Kornev & Taylor, *TiBS* 2015.

5. On a related subject, could the authors insert a brief comment on page 9/10 to describe how the conformation of TNIK in their structures compares to that of the model in the PDB (2X7F)?

6. Page 13 - The authors discuss a structural basis for the difference in cellular activity between NCB-0846 and Compound 9 in terms of the scaffold domain of the kinase. To my knowledge, there is no data on the relationship between the conformation of the kinase domain of TNIK and its scaffolding function and so this is purely speculative. The authors should modify the text accordingly. This would be a more convincing argument if the description of the kinase conformation and corresponding figure were improved (see point 4 above).

Replies to Reviewers

Reviewer #1 (Remarks to the Author):

Tesshi Yamada and colleagues present here an interesting paper, identifying a new orally available TNIK inhibitor, NCB-0846. The authors first show TNIK deficient mice to be resistant to Azoxymethane induced tumorigenesis. In a screen they identify NCB-0846 and its diastereomer NCB-0970. They test the efficacy of NCB-0846 in inhibiting cancer cell growth both in-vitro and in-vivo. Exploring the structural basis of TNIK inhibition, they discover that NCB-0846 binds to the inactive TNIK-KD which in turn confers Wnt signal -inhibitory effects. Finally using 2 mouse models, they evaluate the anti-tumor activity of NCB-0846.

Thank you very much for your constructive evaluation. We have revised the manuscript in line with your comments as follows:

The novelty of the paper lies in the identification of a new and potent Wnt signaling inhibitor NCB-0846, which could prove to be a promising therapeutic approach. However some of the data presented here are incomplete and need to be improved/repeated to

validate the claims. While statistics have been used, quantification and details are needed in main figures and text.

Major revision:

1) The authors should provide more substantial data for the efficacy of the TNIK inhibitor in colon tumorigenesis. The data presented on Page 8 is insufficient and should either be repeated or details need to be added.

We have revised the sentences describing the efficacy of the TNIK inhibitor against colon tumorigenesis (Page 8 line 177 to Page 9 line 180) and included the complete quantitative data (the exact number of tumors that developed in each mouse and the size of each tumor) in Figure 3g.

2) The authors do not consider the off-target effects of NCB-0846. For the identified TNIK inhibitor to be considered as a therapeutic, off-target effects on other signaling pathways need to be ruled out. A gene expression profiling to identify these pathways might be beneficial.

It is known that many signaling pathways are regulated by protein phosphorylation. Therefore, gene expression profiling is not a suitable approach for characterization of protein kinase inhibitors. In our experience, reverse-phase protein array (RPPA) is the most potent method for profiling phosphorylated proteins (Masuda et al., *Biochim Biophys Acta* 2015;1854:651-7). In fact, we have successfully identified aberrant activation of the mTOR pathway in hepatocellular carcinoma resistant to a kinase inhibitor, sorafenib, using our original high-density RPPA (Masuda et al., *Mol Cell Proteomics* 2014;13:1429-38).

Using RPPA, we are now profiling the phosphorylation status of signaling proteins in colorectal cancer cells treated with NCB-0846. RPPA is an unautomated time-consuming method and requires extensive validation. We intend to report our results and their biological implications in our next paper.

3) It is not appropriate to move the whole methods section to the supplementary.

Due to the tight word limitation of *Nature Medicine* (< 3000 words), to which we originally submitted the manuscript, we had to move the whole Methods section to Supplementary materials. In the revised manuscript, we have included details of some specific experiments in the Methods section (Page 15 line 336 to Page 18 line 403).

Minor revision:

1) Page 6, Line 7 - *Quantification is required. 'Small number of incidentally developed large tumors' is vague. Numbers need to be provided and added to text and figures.*

We agree with your comment. We have deleted the vague sentence and included a complete set of quantitative data (the exact number of tumors that developed in each mouse and the size of each tumor) in Figure 1e.

2) Figure 1h- *Can the authors comment on why the tumors increase in size in the colon? Quantification is required (how many were screened and how many had large tumors).*

We have included a complete set of quantitative data (the exact number of tumors that developed in each mouse and the size of each tumor) in Figure 1h. We reconfirmed that there was no significant increase in the size distribution of colon tumors in mice treated with NCB-0846 (Fig. 1h, right).

3) Page 8, Line 22 / Figure 3g- *Data on colon is not satisfactory and needs to be improved.*

We have revised the sentences describing the efficacy of the TNIK inhibitor against colon tumorigenesis (Page 8 line 177 to Page 9 line 180) and included a complete set of quantitative data in Figure 3g. We reconfirmed that there was no statistically significant increase in the size distribution of colon tumors in *Tnik*-deficient (-/-) mice treated with AOM (Fig. 1e, right).

4) *Supplementary S4 could be improved.*

Thank you for this suggestion. The microscopy images have now been revised with the help of an experienced pathologist (Supplementary Fig. S4).

5) *While mouse models provide useful insights, the intestinal organoid technology provides a robust system for such drug efficacy studies. We suggest that the authors also test the potency of NCB-0846 on human tumor organoids (colon and small intestine).*

Thank you for this suggestion. We are now trying to establish organoid cultures of colon cancer cells derived from patients. We would like to report the effects of NCB-0846 on such human tumor organoids elsewhere.

Reviewer #2 (Remarks to the Author):

The manuscript written by Yamada and colleagues describes an extensive and thorough programme of research on the development and characterization of ATP-competitive

TNIK kinase inhibitors. In my opinion, the study is well presented and interesting. There are aspects of it that could be improved, and I would ask the authors to respond to the following points:

1. Figure 1b -this data is not described clearly in the text - the sentence referring to this panel describes the weight of mice, but does not mention the PCR results or the expression levels of TNIK (which is what the data shows). Could the authors also briefly explain their strategy (why an exon 4 deletion)?

Thank you for pointing out the unclear nature of our description. Due to the tight word limitation for our original submission, we had to omit a detailed description of the experiments shown in Figure 1b. We have now added new sentences describing the results of these experiments (Page 5 lines 92-94). The targeting construct was designed to remove almost the entire kinase domain of *Tnik* including the critical ATP-binding site (Page 5 lines 90-92).

2. Figure 2 - Could the authors comment on the apparently poor translation of in vitro potency into one of their cell-based assay (Fig. 2d)? Based on the immunoblot of anti-phosphoserine, the cellular IC50 appears to be in low micromolar range, about 100-fold higher than in vitro potency. This doesn't fit with much of the data that follows in subsequent figures, and it's important to resolve exactly what the potency of the inhibitor is against the target in cells.

The TCF4 substrate protein has several phosphorylation sites. Although complete dephosphorylation of TCF4 required at least 3 μ M NCB-0846, the inclusion of 0.1-0.3 μ M NCB-0846 was sufficient to cause faster migration of the TCF4 protein (Fig. 2d), indicating the dephosphorylation of one or more sites. This concentration is comparable with the active range of NCB-0846 in cell-based assays. We have added new sentences describing this (Page 7 lines 142-145).

3. Figure 2e - Is TNIK autophosphorylated in cells (reference), and why is loss of phosphorylation not observed after 4 hours?

At present, we do not know the molecular mechanisms responsible for the slow inhibition of TNIK phosphorylation. We intend to investigate these mechanisms further.

4. Page 9/10 - The conformation of TNIK in apo form or bound to NCB-0846 is described as inactive, despite having DFG-in and C-helix in positions. This is confusing, and probably inaccurate, because the structure figures show differences in the DFG and C-helix in the structure of TNIK (bound to Compound 9) that looks like a truly active

conformation. I think it would help here to include a figure panel showing the positions of these structural features compared to a well-known kinase in its active conformation such as PKA. Moreover, it would be useful to show the hydrophobic R-spine residues in these structures - this is a recent, and now widely accepted, concept and, based on the available images, it looks like there might well be a difference that would help to clarify whether the TNIK structures are indeed active or inactive. Then the authors we be able to write a more accurate description of th e structures. For information on the R-spine , see Kornev & Taylor, TiBS 2015.

We are grateful to the reviewer f or this comment, which has helped us to im prove the manuscript. In accordance with the reviewer's suggestion, we have removed the former inaccurate description and added a description of the structural features related to R-spine residues (Fig. 4a, 4b, and 4e has been changed accordingly) (Page 10 lines 208-210, Page 26 lines 633-635, and P age 27 lines 641-645). We have also added a new figure panel (Fig. 4f) to allow comparison with a well-known active conformation of PKA (Page 10 lines 217-220 and Page 27 lines 646-647). W e have also added a reference (Kornev & Taylor, TiBS 2015) giving information on the R-spine (Page 21 lines 489-490).

5. On a related subject, could the authors insert a brief comment on page 9/10 to describe how the conformation of TNIK in their structur es compares to that of the model in the PDB (2X7F)?

In accordance with the reviewer 's suggestion, we have added a description to allow comparison with the pre viously deposited TNIK structure. Briefly, the previous 2 X7F model resembles our Compound 9-bound TNIK in the active conformation (Page 10 lines 211-213).

6. Page 13 - The authors discuss a structural basis for the difference in cellular activity between NCB-0846 and Compound 9 in terms of the scaffold domain of the kinase. To my knowledge, there is no data on the r elationship between the conformation of the kinase domain of TNIK and its scaffolding function and so this is puely speculative. The authors should modify the text accor dingly. This would be a mor e convincing argument if the description of the kinase c onformation and corr esponding figure were improved (see point 4 above).

We agree that the conformation of the kinase domain of TNIK and its scaffolding function is purely speculative. Therefore, we have changed the description accordingly (Page 14 lines 306-309).

Finally, on behalf of all co-authors, I thank the reviewers again for their constructive and valuable comments regarding our manuscript. I believe that we have appropriately addressed all the points raised, and hope that the revised manuscript is now suitable for publication in *Nature Communications*.

Sincerely yours,

Tesshi Yamada, MD., PhD.
Chair and Chief, Division of Chemotherapy and Clinical Research
National Cancer Center Research Institute
5-1-1 Tsukiji, Chuo-ku, Tokyo 104-0045, Japan
Fax: (81)-3-3547-6045
E-mail: tyamada@ncc.go.jp

Reviewers' comments:

Reviewer #1 (Remarks to the Author):

In this revised submission, Tessi Yamada and colleagues have addressed most of the concerns raised and revised the text and figures accordingly. Overall the manuscript has improved.

While they added details and quantifications for their experiments on the efficacy of TNIK inhibitor in colon tumorigenesis (Figure 3g), they did not perform any further experiments to improve their results. Furthermore, to address the concern about off-target effects, they intend to perform RPPA. It would be important to include these data in this manuscript or report them in subsequent studies to further validate their conclusions on the TNIK inhibitor NCB-0846.

Reviewer #2 (Remarks to the Author):

The authors have addressed my concerns with the previous version of the manuscript and I am happy to recommend publication.

Replies to Reviewers

Reviewer #1 (Remarks to the Author):

In this revised submission, Tesshi Yamada and colleagues have addressed most of the concerns raised and revised the text and figures accordingly. Overall the manuscript has improved.

While they added details and quantifications for their experiments on the efficacy of TNK inhibitor in colon tumorigenesis (Figure 3g), they did not perform any further experiments to improve their results. Furthermore, to address the concern about off-target effects, they intend to perform RPPA. It would be important to include these data in this manuscript or report them in subsequent studies to further validate their conclusions on the TNK inhibitor NCB-0846.

Thank you very much for your favorable evaluation. Using RPPA, we are now profiling the phosphorylation status of signaling proteins in colorectal cancer cells treated with NCB-0846. RPPA requires extensive validation by immunoblotting. We thus would like to report the results in a subsequent study.

We thank the reviewers again for their constructive and valuable comments regarding our

manuscript. I hope that the manuscript is suitable for publication in *Nature Communications*.

Sincerely yours,

Tesshi Yamada, MD., PhD.
Chair and Chief, Division of Chemotherapy and Clinical Research
National Cancer Center Research Institute
5-1-1 Tsukiji, Chuo-ku, Tokyo 104-0045, Japan
Fax: (81)-3-3547-6045
E-mail: tyamada@ncc.go.jp